# Co-Creating ICT Risk Strategies with Older Australians: A Workshop Model

**DOI:** 10.3390/ijerph20010052

**Published:** 2022-12-21

**Authors:** Jacob Sheahan, Larissa Hjorth, Bernardo Figueiredo, Diane M. Martin, Mike Reid, Torgeir Aleti, Mark Buschgens

**Affiliations:** 1School of Design, College of Design and Social Context, RMIT University, GPO Box 2476, Melbourne, VIC 3001, Australia; 2School of Media and Communications, College of Design and Social Context, RMIT University, GPO Box 2476, Melbourne, VIC 3001, Australia; 3School of Economics, Finance and Marketing, College of Business, RMIT University, GPO Box 2476, Melbourne, VIC 3001, Australia; 4School of Marketing and Management, University of Waikato, Hamilton 3240, New Zealand

**Keywords:** older adults, ICT risks, digital literacy, scenario personarrative method

## Abstract

As digital inclusion becomes a growing indicator of wellbeing in later life, the ability to understand older adults’ preferences for information and communication technologies (ICTs) and develop strategies to support their digital literacy is critical. The barriers older adults face include their perceived ICT risks and capacity to learn. Complexities, including ICT environmental stressors and societal norms, may require concerted engagement with older adults to achieve higher digital literacy competencies. This article describes the results of a series of co-design workshops to develop strategies for increased ICT competencies and reduced perceived risks among older adults. Engaging older Australians in three in-person workshops (each workshop consisting of 15 people), this study adapted the “Scenario Personarrative Method” to illustrate the experiences of people with technology and rich pictures of the strategies seniors employ. Through the enrichment of low-to-high-digital-literacy personas and mapping workshop participant responses to several scenarios, the workshops contextualized the different opportunities and barriers seniors may face, offering a useful approach toward collaborative strategy development. We argued that in using co-designed persona methods, scholars can develop more nuance in generating ICT risk strategies that are built with and for older adults. By allowing risks to be contextualized through this approach, we illustrated the novelty of adapting the Scenario Personarrative Method to provide insights into perceived barriers and to build skills, motivations, and strategies toward enhancing digital literacy.

## 1. Introduction

The growth in the use and adoption of information and communication technologies (ICTs) by older adults—accelerated by the COVID-19 pandemic—provided unique opportunities and challenges. Most explicit during the pandemic was how technologies became essential to reducing social isolation and related issues [1]. This trend, alongside a broader dependence on devices, reflects older adults’ use of ICT to stay and age in place (i.e., at home) longer, support decision-making, and access and share information [2]. The movement to “digital by default” inadvertently left many older adults behind [3]. An outcome of these circumstances is that an individual’s capacity to learn, use, and adapt to technology (i.e., their digital literacy) is inextricably linked to their wellbeing and quality of life [4].

A lack of exposure and experience with ICT can lead to hesitancy around technology use. Research demonstrates how personal circumstances, such as marital status and residential status, can influence device support and digital inclusion for older adults [5]. The likelihood of ICT take-up and use can also be informed by attitudes, education level, and past occupation, with perceptions of technology cost and cognitive ability limiting interest [6]. The literature highlights that those in later life do not have homogeneous device adoption and technology use; as a cohort, older adults are one of the most divergent segments on digital literacy spectrums [7,8]. Training remains a key feature of successful ICT adoption. Learning is most successful when it is linked to real-world needs, cooperative and informal, and supported by memory aids, which are essential to developing strategies and richer experiences [9]. The aim of this collaborative endeavor with older adults was to understand how those in later life manage their issues and the complexity of ICT use in their day-to-day lives.

In this study, we explored the ways perceptions of ICT risks shape older adults’ use and how co-created strategies could support individuals and organizations to overcome barriers. The tailoring of strategies to support ICT adoption and refining information delivery provides an opportunity to improve digital literacy beyond existing educational models. This study focused on the following question: How can we support older adults’ digital literacy by understanding their perceptions of risk and developing strategies with them? Centering on approaches to this issue, this study examined how participatory methods, which is a maturing approach to aging research, can support and enable access to a digitally engaged society.

The study engaged the Scenario Personarrative method [10] as a novel approach to collaborating with older adults and developing nuanced and contextualized ICT engagement strategies. Section 2 presents further literature on the role of perceived ICT risks in limiting the digital literacy of older adults and the role of human-centeredness in technology design. Section 3 provides an overview of the workshop methodology, detailing the development of persona and scenario methods and the adaption of the scenario personarrative method to the context of technology and later life. Section 4 frames the perceived risks and potential strategies around the three degrees of digital literacy that the persona enrichment explored. Section 5 explores the implications of these findings and the benefit of this methodology in addressing the research question. Finally, Section 6 includes a summary of the findings and implications for future studies.

## 2. Related Work

Examining the role of perceived risks and co-designed strategies requires an understanding of how digital literacy shapes the experiences of older adults. Due to the scale of the concept, digital literacy does not have a universally agreed-upon definition. However, it does encompass a range of technical, cognitive, and socio-emotional aspects of becoming competent with a computer [5]. Within our context, improving digital literacy for older adults can vary widely from getting set up with a device to becoming a “power user”. The variety of digital literacy skills and changes in those skills reflect an individual’s experiences of everyday ICT use [11]. While digital literacy upskilling programs may seek to teach technologies prescriptively, scholars have emphasized that these digital workshops should employ more flexible culturally and socially sensitive practices toward building trust and confidence rather than a pre-destined outcome [4,12]. With these practices in mind, rather than focusing on digital literacy through learning and education [13], we noticed that perceptions of ICT risk became a central concern, as the COVID-19 pandemic disrupted the regular technology classroom setting and instruction for older adults rapidly moved online.

Perceived risks reflect older adults’ genuine concerns about the potential harm or the possibility of loss that technology can often pose. These risks may be security- or privacy-related, crossing many aspects of daily life—from financial concerns, performance issues, physical injury, social harm, time-related losses, and psychological risks [14]. While the pandemic has undoubtedly highlighted the perceived usefulness of ICT for social connection [15], human–computer interaction (HCI) researchers noted and demonstrated the barriers to adoption and use. These barriers include perceived surveillance and paternalism from children; explicit barriers, such as physical abilities or financial capability; and structural misperceptions or stigmatization of older adults remain [16]. Scholars from gerontology, psychology, and human behavior studies point to the role of training in developing confidence, indicating the importance of family members and caregivers empowering the use of and exposure to ICT [15,17,18]. While these recommendations are useful for those supporting older adults, they lack an understanding of older adults’ experiences with the ICT strategies they employ.

The participation of older adults and inclusion in technology development is well-established; however, efforts to engage how they appropriate and adapt ICT devices are less so. Studies that did involve older adults in evaluating emerging technologies for older adults, such as virtual reality (VR), focused on how participatory methods can help to build a stronger collaborative partnership, highlighting the benefit of engaging individuals with vastly different levels of technical knowledge [19]. While product-orientated research does offer important insights into design and development, such as how wearable technology can be perceived to diminish an older user’s independence instead of improving it [20], increasingly, the actions of the users in domesticating products and services are being highlighted [21]. As socio-gerontechnologist Alexander Peine [21] suggested, the agency of older people in relation to technology is better explored through supporting these individuals to become co-creators and informed users, as well as reframing perceptions of them as “passive recipients of technology.” If digital literacy centers upon making individuals more competent, supporting more active roles and strategizing with older adults can greatly improve their outcomes with ICT use. If existing interventionist approaches to technology in later life are limiting in realizing digital practices, modes of research that support self-management and self-care can address issues of technology-driven neglect and burdens for older adults.

This study offered an analysis of a series of collaborative workshops conducted with older adults to support their forming and sharing of strategies and knowledge around reducing their perceived risks of ICT. The workshops centered on their perceptions of ICT aspirations, barriers, and strategies for older adults with various digital literacies. Conducted with older Australians (65 years and older), the workshops engaged the senior residents of several local government areas who collaborated on two activities: persona enrichment and scenario mapping. Adapting the scenario personarrative method [22], these activities centered around a spectrum of low-to-high-digital-literacy personas, which participants enriched before mapping their persona responses to several scenarios, indicating how they would cope with ICT failure or seek support. Below, we detail our adaptation of this existing method to suit an aging population in a technological context and describe the accessibility and flexibility the method offers in the context of the pandemic. These aspects were central to how these workshops provided a platform for encouraging new discourses around digital literacy through the more human-centered lens of ICT risk and strategies.

## 3. Methods and Materials

### 3.1. Scenario Personarrative Method/Workshopping Approach

The series of workshops, which took place from January to April 2022, was based on a collaborative “Scenario Personarrative method” [10], which effectively engages with the multi-dimensional factors of digital literacy. This method combines two tasks—scenario thinking and persona narratives—in a common participative approach that Flore Vallet and colleagues [10] saw as a structured yet flexible process. As they highlighted, scenarios deal with broader trends, using projections to build several narratives that detail alternative possible futures to navigate through. Whilst personas may draw on demographic data or not, they often form highly detailed yet representative profiles that express the social and political aspects that technology designers might not consider.

This combination of two often discrete activities offers many benefits, such as enabling “multiple views of an interaction” [23], with this approach enabling “participants to use their own experiences and perceptions to help guide the development of complex interventions” [24]. This fosters a very different type of information than focus groups or usability walkthroughs can, as persona and scenario activities engage with the lived experiences of people, such as those with certain disabilities and health conditions [22]. When used for primary care interventions [24], speculating on autonomous technologies [25], or disaster responses [26], this combination of persona–scenario exercises has seen participants develop fictitious yet authentic personas placed into several scenarios that participants must work together to understand how they might navigate it [27]. These qualities illustrate how this approach can be valuable in the context of eliciting ICT strategies from older adults by navigating perceived risks with them.

More broadly, workshops form a common method in collaborative design research, supporting the evaluation of propositions and co-creation of innovations that we have employed to capture older adults’ thoughts and responses. As a participatory approach where individuals can learn and creatively problem-solve together, workshopping focuses on how participants can express themselves through different channels by facilitating and capturing discussion and constructive processes [28]. In essence, workshops provide a clear purpose and set of goals to participants, involve pre-allocated roles, and often incorporate various research methods ranging from group discussion to creative tasks to interviews and observations [29].

As detailed in Figure 1, the workshops this study focused on were centered around understanding older adults’ perceived ICT risks and strategies in relation to their digital literacy. For the workshops, personas were developed that represented several older adults that were defined by digital literacies ranging from low to high. In addition, the scenarios crafted were situations of potential or actual risk, where ICT strategies that were appropriate to the persona in question needed to be applied. These sessions engaged older adults as collaborators whose activities and discussions were facilitated by a member of the research team. Using printed templates and sticky notes to capture and respond, groups worked through the persona enrichment and scenario mapping exercises. Each activity was supplemented by a wider workshop session involving discussion and reflection on group responses. We note how sticky notes can be used to record, visualize, and communicate emerging insights as an effective method of data collection [30]. The resulting analysis of these materials, enriched personas, and mapped scenarios formed the key results in this article, as arranged by the fictional senior’s digital literacy.

### 3.2. Persona Enrichment Activity

Serving as the first activity and co-design task, groups were allocated a basic persona to enrich with background, motives, and life based on their own experiences and perceptions of individuals with these qualities. The basic persona did not have a name, only their age; brief biography; and an indicative measure of digital literacy skills, their perceived risk of technology, and general health (Figure 2). A persona would be allotted to a group and introduced by the facilitator, with questions around certain concepts and terminology explained and discussed in the group (for example, digital literacy would be introduced as the competency of someone with a given technology; however, the group might decide that speaks more to their skill in navigating an interface than being knowledgeable with the programming of a device). After the group reviewed and discussed this individual, their facilitator would then direct participants to “enrich” this basic profile, offering several prompts for them to respond to. These prompts would be listed on a large, printed template (Figure 3), with sticky notes, markers, and other stationery provided to capture thoughts and comments. These prompts focused on the potential aspirations and barriers for this persona, deciding their sources of information, the types of technology they may own, and the health priorities this persona might have.

These prompts and the supplied materials could further “enrich” this simple persona with life experiences, habits, and behaviors, as well as the concerns and skills they might bring to the following scenario mapping. Provided with an hour to complete the templated grid of prompts, each group would provide aspects and qualities via single-word-to-sentence-long responses on sticky notes. At the end of the session, an elected speaker for the group spoke to the wider room about their enriched persona, describing the narratives they had created around this individual, such as their background or perspective on ICT use, alongside explaining the group’s thinking or any other reasoning. Each group’s persona could then be used in the following scenario mapping, providing in-depth and well-understood characters for participants to navigate the ICT risk situations.

### 3.3. Scenario Mapping Activity

Following the persona enrichment, the scenario mapping involved the same groups placing their persona into three different scenarios, mapping out the problems they might face and the solutions they might employ. Each of the scenarios was based around a common ICT risk that the researchers had earlier documented in a corresponding survey [31], with the first being “Not Getting it”, relating to operational and functional risk; the second being “Hidden Costs” to personal and social risk; and the third being “Purchases Online”, relating to purchase transaction risk. Each contained a short description of the scenario to respond to (Figure 4) through a sheet template with columns concentrated on the persona’s “Problem they will face” and “Solutions they might use”. Like the persona enrichment activity, participants worked in groups with a facilitator and were provided an hour to detail the issues facing their persona, detail the various ways they might address them, and demonstrate strategies.

### 3.4. Participants

The three workshops involved a total of forty-seven participants, with a one-day event conducted in northern Melbourne at community centers in Wollert (n = 13) and Thomastown (n = 25, n = 9). The participants at each session were a combination of thirty-nine older adults and eight council staff from the local area. The workshop volunteers were recruited via advertisements at a local University of the Third Age, council newsletters, and word-of-mouth in coordination with local council leadership. Participants were invited to review the study information and book a session to attend via an online system, with the bookings used to determine the number of facilitators at each session. At the sessions, each participant was then assigned a prepared table with one facilitator and three to five other participants (Figure 5).

### 3.5. Ethical Considerations

Approval for this study was organized with RMIT University’s Business and Law College Human Ethics Advisory Network, which stipulated the recruitment methods, the format of written consent, and the data to be collected for analysis. After being recruited, attendees reviewed the study information and booked a session. They were provided formal consent documents on arrival alongside a workshop overview presentation, with the opportunity to ask questions. Written consent was collected before the activities commenced, with a gift card offered to attendees at the end.

### 3.6. Data Synthesis and Analysis

The data from the workshop formed a series of templates filled with sticky notes containing participant responses, which were processed through qualitative content analysis. This approach reflects efforts to code qualitative data from sticky notes that are typical in participant-based concept mapping, such as that conducted here [30,32]. Intended to help organize and elicit meaning from the data collected, content analysis makes inferences from texts, forming realistic conclusions alongside describing specific phenomena from such datasets [33]. Following Mariette Bengtsson’s [33] process for the content analysis process, the data this study examined were synthesized systematically, drawing on deductive reasoning to address the predetermined and defined topics as a coding list (i.e., ICT aspirations and barriers, health priorities, etc.). With the individual sticky notes forming the study’s units of analysis, the researchers familiarized themselves with the data by transcribing the notes before deductively coding the resulting texts and locating and grouping data. These initial rounds of coding saw broad descriptors attached to each of the six categories, which we then sorted around the corresponding levels of digital literacy attached to each. These lengthy and detailed descriptors were then condensed into sub-themes that articulated the original content, as discussed below.

## 4. Results

### 4.1. High-Literacy Persona

When enriching higher literacy personas, the participants focussed on how their level of skill provided a platform for expanding their activities and bringing multipurpose characteristics to the fore (Figure 6). Personas leveraged their devices to gain accessibility, gather information, search/plan for things, enhance their hobbies, or support others. While this contrasted with the lower levels, the barriers appeared more closely aligned with these personas dealing with their own limitations, such as patience to learn and the high cost of technologies, as well as being limited by their specific location, with regional personas having lower online access and device availability than their lower digital literacy counterparts. There were also similarities with where they might go for support because even if they were more skillful/knowledgeable, participants discussed often needing in-person support frequently, though this was supported with easier access via Google searches and YouTube. The prevalence of technology in the lives of these personas was very apparent, with devices used to communicate (smartphone, etc.), browse and watch content (iPad, laptop, smart TV), and track health (FitBit), with the use of tech being constant and everywhere also posing questions around addiction and overuse potential.

### 4.2. Medium-Literacy Persona

Whereas we saw a multiplicity of aspirations of higher-level digital literacy personas, the medium-level persona had an increasingly functional set of aspirations (Figure 7). These were set around increasing social connection and becoming more comfortable with using devices for everyday things like paying a bill or using professional software. However, like the high DL persona, participants documented how barriers formed around the mindset and level of experience, with an apparent lack of confidence combined with a lack of assistance being evident limiters. Issues due to the cost of technology were similar, but concerns around their terminology and concepts for searching/understanding were less so. Participants in the co-design sessions highlighted that an individual of this level would seek multiple, localized sources of support and draw on “call-a-geek” services, though they were mindful of the cost. Several noted the importance of language-based support and peer groups for culturally and linguistically diverse personas at this level.

Device use featured prominently in the home, with mobile and portable devices being essential “on-the-go” ICT. This ICT use was focused on social connection with grandchildren, friends, and groups, as well as booking check-ups with the doctor. This saw a clear set of devices on hand that responded to either play (media, entertainment, gaming, etc.) or focus (administration, travel, professional work, etc.) contexts. For example, desktop computers, mobile phones, and GPS units were essential to completing important daily tasks. In contrast, an iPad or smart TV would be used for infrequent gaming or entertainment purposes, with some participants indicating their persona might aspire to involve a wearable smartwatch in their routine to capture health data.

### 4.3. Low-Literacy Persona

Lower-digital-literacy personas, in comparison to medium-level ones, saw several severe limitations on their ability, which was reflected in the aspirations and barriers they had (Figure 8). Like the medium-level personas, they were primarily concerned with using technology to connect and socialize, mainly calling family, viewing updates via social media, and getting support in emergencies (having a fall, calling 000, etc.). This reflected the barrier to technology use, perhaps because they were limited due to low health or a lack of interest in devices that prioritized practical applications. At the same time, many suggested these individuals might not be easily pressured into using any or all ICT. This resistance was characterized as a further barrier, which some participants attributed to personal worries around memory loss and an inability to learn. As such, support for devices came from their immediate surroundings, such as nurses and carers, and reaching out to family members, local council, and social workers. Use would arise when issues or emergencies occurred due to owning a mobile and home phone to receive messages and having access to second-hand tablets or computers.

This approach to clustering the dataset provides a useful way of capturing the risks and responding strategies by separating them into perceived degrees of skill, illustrating how older adults can view their peers’ digital literacies. As discussed, previous studies documented the individual and environmental factors that shape the perceptions of risk and skills an older adult can hold, but not explicitly with those older adults. By employing personas with diverse skills to capture these factors, we detailed how older adults attribute common limitations and mindsets to each level of digital literacy, indicating that these levels might be effective at realizing and responding to an individual’s needs. The benefit of the workshop setting is apparent in how we noted these levels were interlinked, as participants suggested how growing confidence and further social opportunities were often reasons that older adults developed their literacies around technology. Exploring digital literacy through the more heterogeneous framing these persona and scenario exercises offer may better contextualize well-established factors for older adults in collaborative settings.

## 5. Discussion

This study focused on how we might support digital literacy in later life by understanding ICT risks and developing strategies. The workshops offered key insights into the potential for adapting scenario personarrative methods to collaborate with older Australians. In drawing on collaborative, co-creative techniques to facilitate this series of workshops, we noted how this approach saw participants actively engage in capturing the aspirations, barriers, sources of support, and ICT use they perceived at varying levels of technical skill and digital literacy. By enriching several fictional personas and mapping how they might respond to risk-based scenarios, individuals applied their own perceptions of ICT risk. They shared their own experiences and strategies with a wider group, enabling discussions that led to three key levels of digital literacy and associated qualities formed through our content analysis.

In describing the contextualized data and nuanced results, we established how adapting this persona–scenario methodology with older stakeholders can support collaborative approaches to understanding the perceptions and strategies that inform digital literacy. We also considered the benefits of this collective experience in helping catalyze practices that addressed the needs and challenges of those participating. Ultimately, we suggested that employing methods that extend the scenario personarrative method can engage a more human-centered agenda. This is particularly true regarding the ICT strategies we supported and the perceived risks we addressed in the design and deployment of technologies (Figure 9).

The qualitative content analysis revealed six predetermined topics categorized by persona digital literacy, establishing key differences and commonalities between older adults of differing competencies online and with ICT. Focusing upon the aspirations of these enriched, fictional individuals, we noted how older adults perceived higher literacy as a platform for becoming a power user, which was centered around ICT experiences and integration rather than the functional and social objectives of medium- and low-level personas. Regarding commonalities, we found multiple barriers associated with all levels of digital literacy. Reflecting the literature, perceptions of aging-related and educational limitations were prominent, while location-based ones, such as the city–country divide, and economic and budgetary restraints were perceived to reduce technology accessibility [16,34,35]. Previous studies generally associated these opportunities and limitations with a combination of individual and environmental qualities, with older adults often highlighting aging-related physical and mental barriers (memory loss, fear, low self-esteem) [34] over less obvious structural (the pandemic, teaching, media skepticism) and implicit barriers (surveillance, paternalism, appliance design) [16].

In responding to the ICT risk scenarios, the sources of support, mindsets, and strategies individuals brought were central to how they addressed the functional, social, or financial problem. Low-to-medium literacies were often perceived as most likely to go to a family member or existing supporter to manage a problem; however, participants suggested that lower-literacy individuals might see a burden in this and limit support over time. In seeing the same burden, medium-literacy individuals could be seen to make these support sessions more educational, drawing on existing methods of recording, listing, and attempting to resolve recurrent issues themselves. Meanwhile, high-literacy personas were most associated with a flexible and problem-solving attitude to these issues, leaving support systems as a backup option rather than the first point of call. Scholars like Alonso Gonzalez and colleagues [7] documented how building on previous skills and seeking informal learning are more engaging for older adults, as evident in the strategies these seniors perceived as being more beneficial to them. This is clear in how those from non-English speaking backgrounds sought community-based ICT support in their own languages rather than just translated materials.

The perceptions of risk and chosen strategies older adults highlighted during the workshops reflected a need for more nuanced and contextualized approaches to supporting their digital literacies. Through workshop discussions and focusing on personas as a way to ground practices with perceptions, we considered the challenges and possibilities for researchers in navigating digital literacy beyond just a pedological paradigm. As indicated, scholars already highlighted how both individual and systematic factors played a role in the capacity of older adults to use and improve their ICT skills and online literacies. In addition to enabling durable skillsets and constructive attitudes, we indicated that interventions and programs acknowledged the diversity and richness of older adults’ digital media engagement spectrum from non-users to tech-savvy users.

In adapting this methodology, we considered how human-centered approaches can effectively engage older adults, the complexity of the issues they face in addressing perceived risks, and the often tacit ways they address them. This combination of persona enrichment and scenario mapping is not only flexible to this context but also reduces the onus on older adults to identify their own literacy and explore that in a group setting, instead applying their perceptions to a fictional individual and problem-solving through them. Because these workshops involved multiple seniors sharing their ideas and strategies through this individual, we saw the potential for these collaborative and informal settings to act as educational and informative opportunities to learn from others (Figure 10). By negotiating contemporary technical risks and social norms with their peers, the workshops involved participants in peer-to-peer discovery and learning that involved finding out about new mobile applications, such as password managers, and better approaches to their issues and local sources of support. While this exercise, in realizing the risks and strategies older adults engage with, can support future programs and interventions, the collaboration also supported participants in discovering new ones, which is a form of mutual learning scholars increasingly document with older adults [36].

These findings suggested we need to offer more nuanced models that acknowledge the diversity and richness of older adults’ digital media engagement spectrum from non-users to tech-savvy users. They also added to the notion that those in later life do not require unique interventions compared with the broader population. Instead, programs, products, and systems that provide more understanding of the contexts, passions, and experiences individuals seek are required. As data further saturates our everyday lives, we will increasingly need creative, co-designed, and speculative ways to think through the possibilities and limits of the digital across various literacies, subjectivities, and spectrums. We need a more robust discussion about the role of methods in workshops to bring often tacit practices and perceptions to the forefront and how those possibilities shape how we move forward together. This is especially true in a data-saturated world in which digital literacy has become synonymous with social inclusion. Approaches such as the scenario personarrative method allow for topics such as perceived risks and ICT use to be discussed in ways that allow for nuance and context. Based on this study, a scenario personarrative approach can help to provide insights into perceived barriers and to encourage older adults to build skills, motivations, and strategies to enhance digital literacy.

The persona enrichment and scenario mapping process described here forms an example of a broader shift in focus from technological to human aspects in the design for later life. As social scientists Linda Nierling and Emma Domínguez-Rué [37] highlighted, this is a move toward questioning how technology is integrated into the social surroundings that shift the focus from technical features to social phenomenon. Persona and scenario exercises, as opposed to usability testing, address this by reshaping the relationship between individuals and technologies, building out the technologies based on a persona and engaging with the barriers and strategies of individuals. In working through several levels of digital literacy, this method also focuses more on human complexity, from personal attitudes to embedded contexts, which is often not engaged within the ICT strategies offered to those around older adults.

Such qualities benefit not only older adults as ICT consumers but also the designers of technologies and educators. Our analysis demonstrated that digital literacy can effectively determine the potential risks associated with an individual and the strategies they use, as long as these aspects are contextualized and examined with the individuals. For both technologists and instructors, this might mean tailoring resources for particular literacies; responding to the aspirations and barriers individuals have; and forming programs, products, or services that better align with the mindset and attitude that level of digital literacy brings. We also hope that in adapting this methodology, and by workshopping these issues with older adults, future studies might refocus on digital literacy beyond the classroom and explore with older adults the dynamics that shape, limit, and enable their ICT use and skills.

## 6. Conclusions

In eliciting the attributes and strategies older adults associated with digital literacy, this study provided further understanding of the factors that inform the use and perceived risk amongst older adults. The series of workshops employed the “Scenario Personarrative Method” to focus on personal experiences and collaborative discourse to enable more nuanced and contextual perspectives on digital literacy in later life. These workshops captured the similarities and differences in how older adults at different levels of digital literacy conceived of their barriers and employed certain strategies in relation to ICT use. This study indicated the promising application of the scenario personarrative method in participatory activities with older adults, indicating the benefits of persona enrichment and scenario mapping for understanding the ICT practices and perspectives of those in later life.

## Figures and Tables

**Figure 1 ijerph-20-00052-f001:**
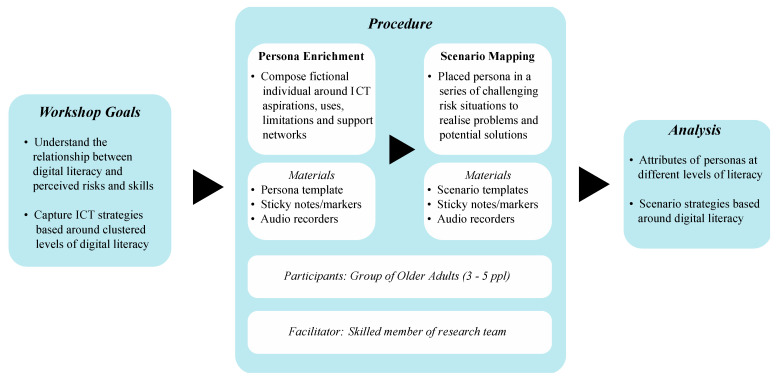
Workshop design.

**Figure 2 ijerph-20-00052-f002:**
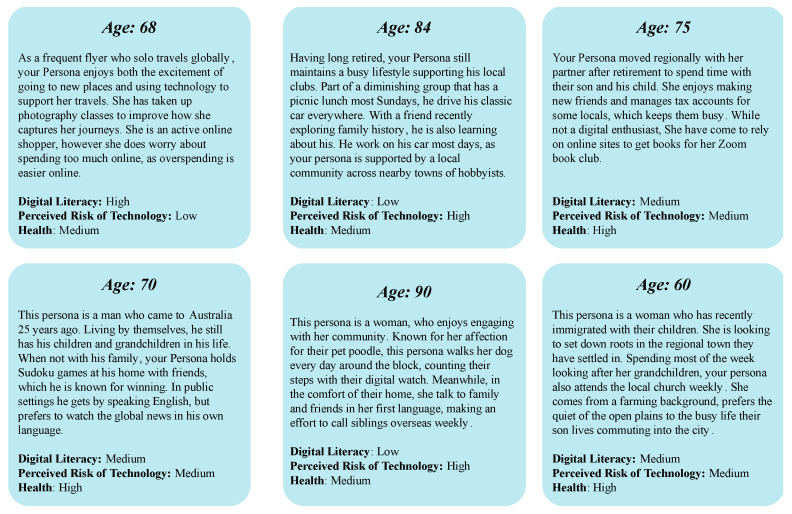
Basic persona profiles.

**Figure 3 ijerph-20-00052-f003:**
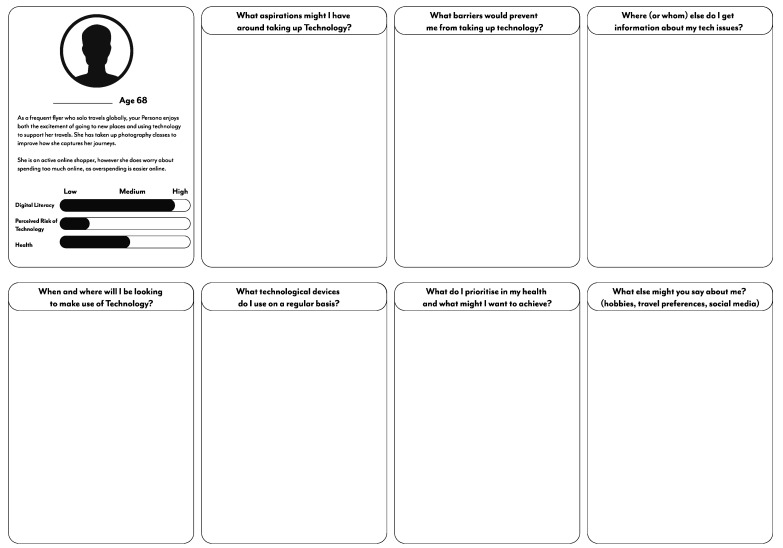
Persona enrichment template.

**Figure 4 ijerph-20-00052-f004:**
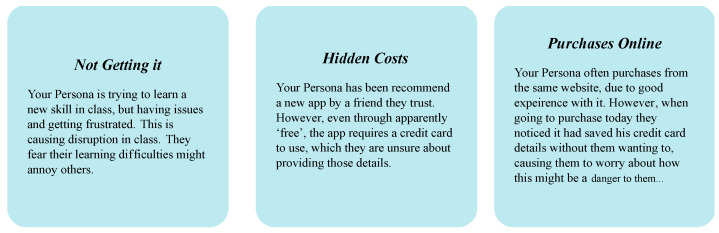
Scenario descriptions.

**Figure 5 ijerph-20-00052-f005:**
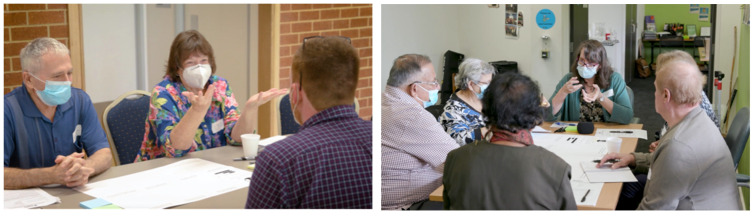
Workshops at the Wollert (**left**) and Thomastown (**right**) locations.

**Figure 6 ijerph-20-00052-f006:**
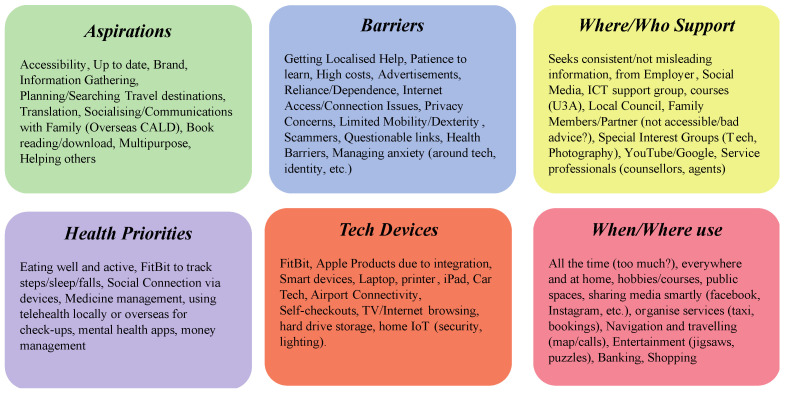
Highly literate persona.

**Figure 7 ijerph-20-00052-f007:**
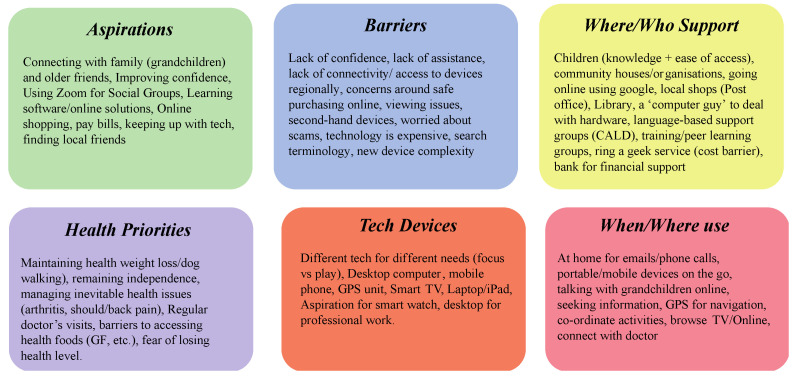
Medium-literacy persona.

**Figure 8 ijerph-20-00052-f008:**
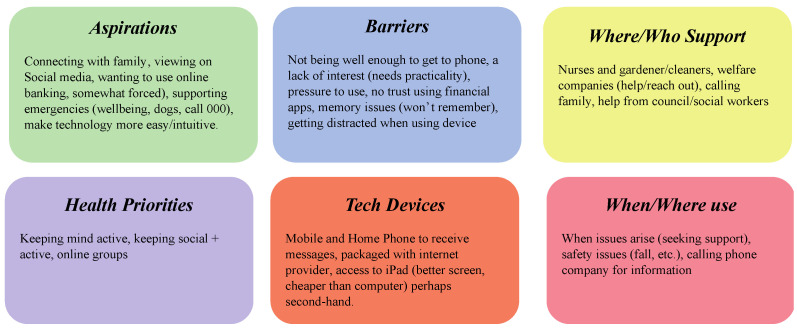
Low-literacy persona.

**Figure 9 ijerph-20-00052-f009:**
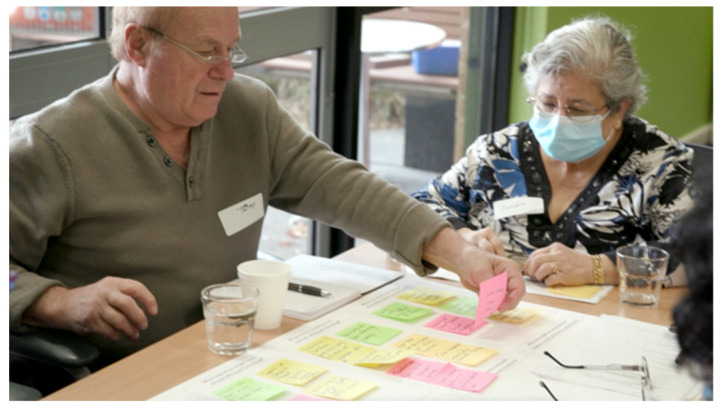
Older adults participating in scenario personarrative activities.

**Figure 10 ijerph-20-00052-f010:**
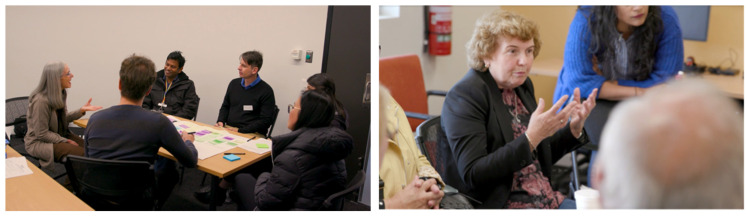
Council staff and older adults contributing to the conversation and agenda.

## Data Availability

The data presented in this study are available on request from the corresponding author. The data are not publicly available due to privacy and ethical restrictions imposed by the Research Ethics Board.

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
