# Peer review of "Co-Creating ICT Risk Strategies with Older Australians: A Workshop Model"

_ijerph, 2022, doi:10.3390/ijerph20010052_

Round 1
Reviewer 1 Report
This paper reports on collaborative workshops with older adults. Using the scenario personarrative method, the authors explored how the older workshop participants perceive other older adults with high, medium or low digital literacy.
The research question is “how do older adults perceive of the ICT aspirations, barriers, and strategies of other older adults based on their digital literacy?” (l.80). Additionally, the authors aimed to investigate how older adults might be supported to form and share their strategies and knowledge around reducing their perceived risks of ICT (l.76). They also “focus upon how the perceptions of risks and barriers to ICT shape older adults’ use, towards exploring how co-created strategies could support individuals and organizations” (l. 43). According to the conclusion (l.312), the authors additionally aimed to use the scenario personarrative method as a way to expand on the discourse on digital literacy in later life by giving space for older adults to share nuanced and contextual perspectives.
I understand that all these aims are important and represent important contributions to the fields of Design, HCI and Gerontology. However, I feel that the paper can benefit from a stronger focus: is it about the method or is it about creating a more detailed understanding of digital literacy. Or is it indeed about both, in which case the authors should add a findings section evaluating the method. I can see merit in all of these, but in its current form, the paper might be too broad. Please also state clearly what the findings are à for example lines 86-88: Our findings point to how perceptions of skill are shaped by an individual’s context, establishing the role of digital literacy, and how this method offers accessible and flexible ways to engage older adults in matters of technology use. This sounds great but raises questions: the role of digital literacy in what? In what way does the method engage older people in technology use? Or rather in discourses around technology?
Literature Review
The literature review of this paper is merged with the introduction section. Even though it broadly covers literature on older adults’ use of ICTs, the main concept underpinning this research, digital literacy, is not explained. I think the paper would benefit from separate introduction and literature sections. This could help strengthen the framing of the paper by stating a clear problem and research approach in the introduction, and then expanding on key concepts in a background literature section. This would also be an opportunity to critically engage with the literature and highlight diverse digital literacy behaviours in later life or engage with other participatory design projects on ageing and technology. This would help strengthen the authors’ (really important!) discussion about diversity and human-centredness in technology design.
Here are some example papers as inspiration:
· Neves, B. B., Waycott, J., & Malta, S. (2018). Old and afraid of new communication technologies? Reconceptualising and contesting the ‘age-based digital divide’. Journal of Sociology, 54(2), 236-248.
· Vines, J., Pritchard, G., Wright, P., Olivier, P., & Brittain, K. (2015). An age-old problem: Examining the discourses of ageing in HCI and strategies for future research. ACM Transactions on Computer-Human Interaction (TOCHI), 22(1), 1-27.
· Grigorovich, A., Kontos, P., Jenkins, A., & Kirkland, S. (2022). Moving toward the promise of participatory engagement of older adults in gerotechnology. The Gerontologist, 62(3), 324-331.
Methods
The method chosen is very interesting and I would like to try it as part of my own research. I have worked with co-created personas before, but for researchers who are not familiar with design methods, please add a description. A short description could be added in the introduction already. In general, this section is lacking detail, especially if the method is one of the main contributions.
Examples:
· First sentence of methods section: What digital literacy profiles are we talking about? What parallels? Were the personas already created and matched with the digital literacy profiles of the participants? What scenarios and events? What ICT strategies?
· line 95: what do you mean by early stage tool and participatory agenda-setting approach? Why did this influence your decision to use this method?
The reader doesn’t have any information about the workshop yet, so it might be good to start the methods section with a general overview of workshops in general, then the method itself and then move into how it was applied in your specific research context. This might help us understand why this method was chosen. Please also add more detail to the method in general (e.g. line 141: the digital literacy of the personas à was there a more theoretical introduction to the workshop to define digital literacy? How did you create a shared understanding of the term digital literacy?). I think the size of the workshops is very impressive and I congratulate the authors for recruiting so many participants in these difficult times. I would also like to know more about the format of the workshops. Did the same participants attend on three days? Or was it 47 different participants each day (making it 141 participants in total)?
Data collection, data analysis and findings
This paper would benefit from a clear section on what the data collected actually encompassed. Just the sticky notes? Why was it video recorded if that was not used as data? This brings up questions around pressure on participants / ethical considerations. The data analysis section also needs a lot more information. What are examples of the coding? Maybe a photo would be helpful? I’m also not sure that reference 26 really fits to describe the process of a content analysis. I looked at the paper and it seems to be more of a comparison between thematic analysis and content analysis, so I would like to read more about why the authors have chosen one over the other if this is the reference used. I think the findings are interesting, but I am unsure what the actual contribution of the analysis was. To me it looks like the authors used the categories (high, medium, low digital literacy) and then the prompts from figure 4 and just mapped the answers. Adding more detail on the process of the analysis is crucial.
Discussion
The discussion section picks up on relevant points, for example the need to diversify our understanding of digitalisation and ageing or highlighting the need for a more robust discussion on the role of methods in workshops. The authors state that those possibilities shape how we move forward together. Indeed, this is exactly what I would like the authors to tell me: What is the role of the scenario personarrative method in shaping research, policy and practice? The authors say that there are key concepts that shape skills, capacity and attitudes (line 273) but what are they? If these are stated more clearly in the findings, they need to be named and expanded on in the discussion. Like the whole paper, this section seems too vague.
Overall, I can see great potential for this paper if the focus is refined and everything is described a lot more clearly so that other researchers can also use the method. I really wish for this paper to be published in the future and I encourage the authors to make the changes. I am not sure if there was a page limit on this paper, but in my opinion adding more detail will strengthen the paper.
Minor edits:
· Figure 2: change the title of the figure, it seems to refer to 3
· figure 1 is nice but maybe misplaced in the introduction
· figures are very hard to read (small font size)
· Please proofread, there are a lot of words missing or misspelled, which could easily be fixed by autocorrect. Unfortunately makes it feel like the authors didn’t put much effort into this part of the paper.
· there are some very long sentences that might benefit from shortening e.g. ll 199-203, 238-241
Author Response
Thank you for your review of our article. Regarding your comments, we have made the following adjustments:
- We have shifted the paper's emphasis onto the methodological contribution, adapting the scenario-personanarrative method. The Abstract and Introduction now reflects this.
- We have separated the literature review into a Related Literature section, which serves to capture scholarship regarding not only risk strategies, but also older adults' roles in technology design
- The methods have been further framed and detailed, with our process of data analysis also more comprehensive
- We have attended to the vagueness described in the Discussion, illustrating more scholarly, policy and methodological benefits from this work.
- Additional minor edits have been addressed
Reviewer 2 Report
Thank you for the opportunity to review this proposal.
The paper approaches an outstanding research topic nowadays, envisaging the adoption of strategies for the digital inclusion of older people. The research approach is very detailed and well-described and consists of qualitative research. The authors conducted three in-person workshops with older adults and adapted Scenario Personarrative Method to develop a digital inclusive scenario.
Some suggestions that might improve the quality of the paper:
- The authors should depict the novelty of their research from the methodological perspective.
- It is not clearly stated what contributions this paper adds to the research or what is the newly generated knowledge.
- Policy recommendations must be framed based not only on theoretical issues but also on the analysis results.
Author Response
Thank you for reviewing our article and your suggestions, we have address them like so:
- The authors have addressed the novelty of our research, and indicated the methodological contribution it makes.
- We have also more clearly stated the contributions and how this paper adds to the research in this space of ageing and technology.
- We appreciated the recommendations regarding policy, and have framed the study's benefits around the analysed results as well.
Reviewer 3 Report
Very interesting paper and experience. For sure you can follow working in this line making more workshops to improve the significance of the activity. Document well structured and defined. The reading is easy to follow and with good rythm.
Author Response
Thankyou for your review of the article, the authors have greatly appreciated the feedback.
Round 2
Reviewer 1 Report
I can see that you have put extensive work into editing this paper and I thank you for considering my previous comments. The paper improved so much. It is now focused and much easier to understand in terms of structure and language, making it much more interesting to the research community. I have no further comments.